# A selenium-catalysed *para*-amination of phenols

Dingyuan Yan[1], Guoqiang Wang[2], Feng Xiong[1], Wei-Yin Sun[1], Zhuangzhi Shi [1],
Yi Lu[1], Shuhua Li [2] & Jing Zhao [1]

Antioxidant enzyme glutathione peroxidase (GPx) decomposes hydroperoxides by utilizing the different redox chemistry of the selenium and sulfur. Here, we report a Se-catalysed *para*-amination of phenols while, in contrast, the reactions with sulfur donors are stoichiometric. A catalytic amount of phenylselenyl bromide smoothly converts *N*-aryloxyacetamides to *N*-acetyl *p*-aminophenols. When the *para* position was substituted (for example, with tyrosine), the dearomatization 4,4-disubstituted cyclodienone products were obtained. A combination of experimental and computational studies was conducted and suggested the weaker Se−N bond plays a key role in the completion of the catalytic cycle. Our method extends the selenium-catalysed processes to the functionalisation of aromatic compounds. Finally, we demonstrated the mild nature of the *para*-amination reaction by generating an AIEgen 2-(2′-hydroxyphenyl)benzothiazole (HBT) product in a fluorogenic fashion in a PBS buffer.

[1] State Key Laboratory of Coordination Chemistry, Institute of Chemistry and BioMedical Sciences, School of Chemistry and Chemical Engineering, School of Life Sciences, Nanjing University, 210093 Nanjing, China. [2] Key Laboratory of Mesoscopic Chemistry of Ministry of Education, Institute of Theoretical and Computational Chemistry, School of Chemistry and Chemical Engineering, Nanjing University, 210093 Nanjing, China. These authors contributed equally: Dingyuan Yan, Guoqiang Wang, Feng Xiong. Correspondence and requests for materials should be addressed to Y.L. (email: luyi@nju.edu.cn) or to S.L. (email: shuhua@nju.edu.cn) or to J.Z. (email: jingzhao@nju.edu.cn)

Selenium is an essential biological trace element discovered by Jöns Jacob Berzelius in 1818 [1]. The selenium analogue of cysteine, known as selenocysteine[2–4] (Sec), is the main biological form of selenium. The most studied selenoenzyme glutathione peroxidase (GPx) has an Sec residue in its active site that is responsible for decomposing hydroperoxides (Fig. 1a)[5,6]. Besides, the flavin-containing redox enzyme thioredoxin reductase (TrxR)[7–9] and the deiodinating enzyme iodothyronine deiodinase (ID)[10,11] represent other key selenium-containing enzymes in biocatalysis.

Selenium-containing small molecules, such as ebselen and its analogues, have also exhibited important antioxidant activity as GPx mimics[12–15]. Organoselenium-catalysed reactions have been widely employed in a number of different reactions[16–18], and substantial progress has been made by Breder[19–21], Wirth[22–24], Denmark[25,26], Yeung[27] and Zhao[28–31] in recent years. Notably,

selenium has emerged as appropriate alternatives to precious metals as catalysts for the construction of C–N bonds[32–34]. Breder et al. discovered an elegant selenium-catalysed amination of allyl and vinyl using N-fluorobenzenesulfonimide as oxidant and nitrogen source[35]. Furthermore, Zhao et al. accomplished a powerful pyridination of 1,3-dienes using (BnSe)$_2$ as a catalyst[36] (Fig. 1b). However, no selenium-catalysed processes for the functionalisation of aromatic compounds have been developed. One challenge might be the electrophilic selenium catalysts react with the aryl rings directly, leading to the deactivation of catalyst[37,38]. We thought that a more nucleophilic site, to accommodate with selenium catalyst temporarily, might be helpful for competing with the deactivation. We herein report a strategy to first form an intermediate with an adjacent, redox versatile Se–N bond which undergoes two successive sigmatropic rearrangements to generate the para-amination product and regenerate the selenium catalyst (Fig. 1c).

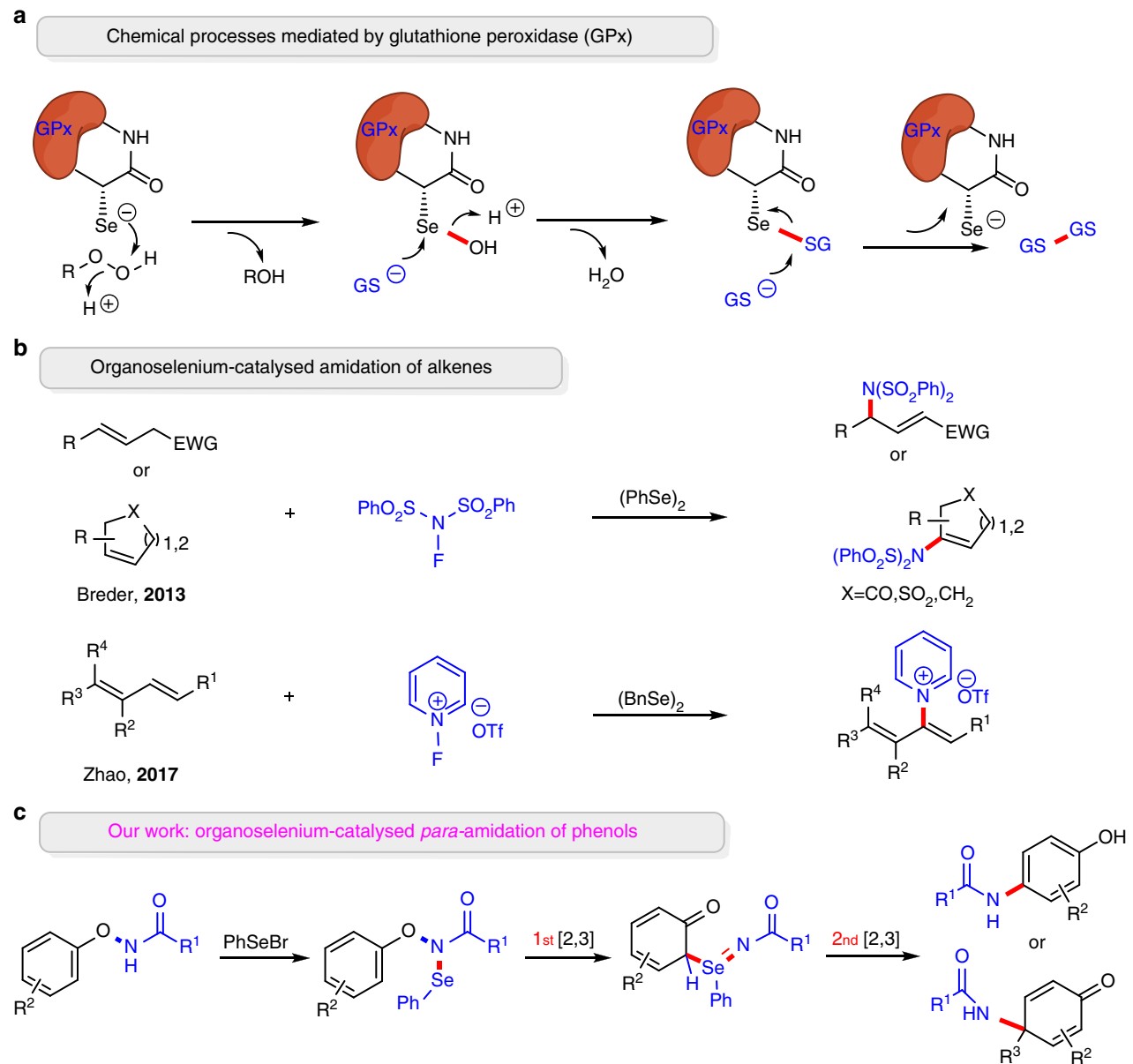

**Fig. 1** Selected biological reaction and organic reactions catalysed by selenium. **a** Proposed catalytic cycle of glutathione peroxidase (GPx) for the reduction of hydroperoxides in biology. **b** Previous reports on organoselenium-catalysed amination of alkenes. GS⁻ glutathione. **c** Our double [2,3]-sigmatropic rearrangement to achieve para-amination of phenols

**Table 1 Substrate scope of Se-catalysed *para*-amination of phenols[a]**

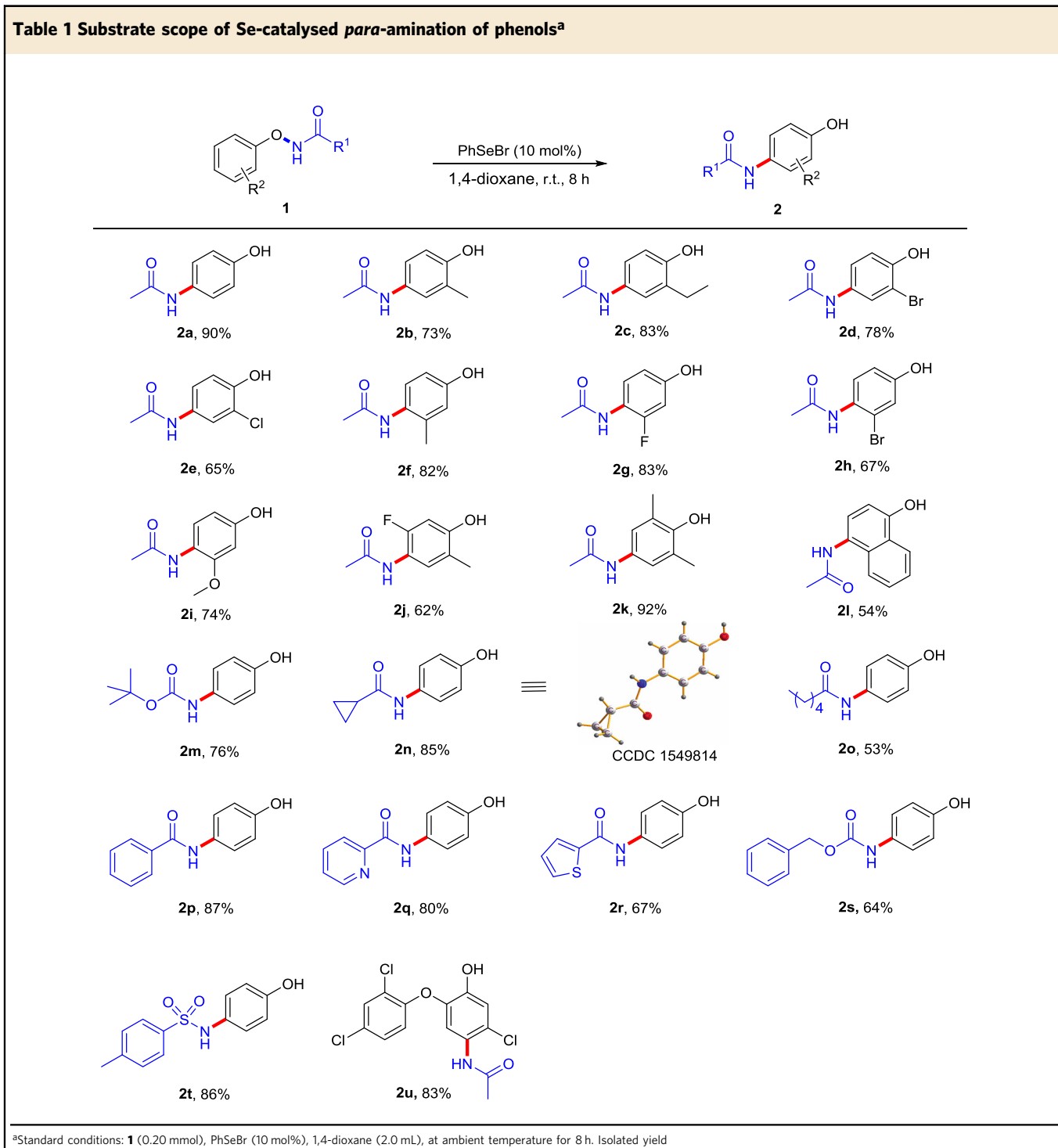

[a]Standard conditions: **1** (0.20 mmol), PhSeBr (10 mol%), 1,4-dioxane (2.0 mL), at ambient temperature for 8 h. Isolated yield

## Results

**Model reactions and substrate scope**. We started by treating *N*-phenoxyacetamide (**1a**) with 1.0 equiv. of *N*-phenylselanylphthalimide (**C1**); we observed the *para*-aminated phenol (**2a**, acetaminophen) in 47% yield. To our delight, when catalytic amount of **C1** (10 mol%) was used, **2a** can be obtained in 38% yield (Supplementary Table 1, entries 1−3). This result compelled us to explore other organoselenium reagents that might catalyse this reaction. No product was detected when diphenyl diselenide (**C2**) and diphenylselane (**C3**) were used (Supplementary Table 1, entries 4 and 5). Both PhSeCl (**C4**) and PhSeBr (**C5**) proved to be efficient catalysts in 2,2,2-trifluoroethanol (TFE) with 60 and 79% yields, respectively (Supplementary Table 1, entries 6 and 7). Screening of a variety of solvents (including MeOH, DMSO, THF, MeCN, EA) indicated that 1,4-dioxane was the best solvent (93% NMR yield and 90% isolated yield of the desired product, Supplementary Table 1, entries 8−13). Ultimately, the optimal reaction conditions employed 10 mol% PhSeBr (**C5**) in 1,4-dioxane at room temperature in air.

**Table 2 Substrate scope of Se-catalysed dearomatization reaction[a]**

[a]Standard conditions: **1** (0.20 mmol), PhSeBr (10 mol%), 1,4-dioxane (2.0 mL), at ambient temperature for 8 h. Isolated yield

The optimized reaction conditions proved to be effective with a number of other substituents on *N*-phenoxyacetamides (Table 1). *N*-phenoxyacetamides with electron-rich or electron-deficient substituents reacted smoothly to give the desired *para*-C–H amination products (**2a**−**k**) in moderate to excellent yields (62 −92%). Electronic effects did not significantly influence the outcomes of the reactions. *N*-phenoxyacetamides bearing fluoro-, bromo-, and chloro-substituents (**2d**−**e**, **2g**−**h**, **2j**) were successfully subjected to this simple protocol with yields from 62 to 83%. The reaction condition was applicable to yield aminated naphthol (**2l**) in 54% yield.

To further expand the scope of this highly *para*-selective amination process, we investigated different *N*-phenoxyamides. *N*-phenoxyamide with the Boc-substituent on nitrogen afforded the corresponding product **2m** in 76% yield. When the acetyl group was replaced by other aliphatic groups such as cyclopropanecarbonyl and hexanoyl groups, the reactions proceeded smoothly to afford the desired products **2n** and **2o** in 85 and 53% yield, respectively. Replacing the acetyl group with aromatic amides or sulfonamide also furnished the desired phenols (**2p**−**t**) in good yields (64−87%). When we applied the method to the late-stage modification of an antifungal drug Triclosan, the desired *para*-aminated product (**2u**) was isolated successfully in 83% yield.

**The oxidative amination/dearomatization reaction**. When the *para*-methyl-substituted substrate was employed, we obtained the dearomatization product **3a** in 78% yield (Table 2). Efficient oxidative amination of phenols was also obtained when ethyl, propyl was present at the *para* site under standard reaction condition. However, we did not detect any of the dienones when methyl was replaced with bulkier substituents, such as isopropyl and *tert*-butyl groups. In those cases, only the corresponding phenols were isolated. Replacing the acetyl group with the propionyl or isobutyryl group on nitrogen gave **3f** and **3g** in 80 and 61% yield, respectively. Finally, protected tyrosine underwent oxidative amination to give **3h** in 56% yield under standard conditions.

**The stoichiometric sulfur-mediated reaction**. The success in the Se-catalysed synthesis of *p*-aminophenols or dienones prompted us to attempt to develop a similar sulfur-catalysed version which could display good catalytic activity as organochalcogen catalysis[39–43]. However, when a solution of **1a** was treated with 10 mol% 2-(*p*-tolylthio)isoindoline-1,3-dione (**4a**) at ambient temperature over a period of 5 h, we detected a trace amount of *para*-aminated product (**5a**) with a preserved N−S bond. When the amount of **4a** was increased to 1.2 equiv., *para*-aminated product (**5a**) was obtained in 38% yield. An extensive screening of bases (e.g. pyridine, CsOAc, 2,6-lutidine, DMAP, Na₂CO₃, DBU, DIPEA) was conducted and revealed that 2,6-lutidine gave the desired *para*-aminated product **5a** in 53% yield. Further optimization established TFE as the best solvent for this transformation, providing the *para*-aminated phenol in 84% isolated yield (Supplementary Table 2).

With the optimal reaction conditions established, we investigated a series of *N*-phenoxyacetamide substrates (Table 3). *Ortho*-substituted *N*-phenoxyacetamides delivered the desired *para*-aminated phenols in good to excellent yields (**5a**–**5c**). When the *N*-protecting group was replaced by other aliphatic amides such as cyclopropanecarbonyl and propionyl groups, the reactions proceeded smoothly to afford desired products **5e** and **5f** in 45 and 80% yield, respectively. The reaction proceeded smoothly with both substrates bearing electron-donating group (**5h**) and halogen-containing *N*-substituted phthalimides (**5i–m**).

**Mechanistic study**. A series of experiments were conducted to probe the reaction mechanism. The *ortho*-sulfiliminyl phenol **5g″** could not transfer to *para*-aminated product **5g** under standard reaction conditions (Supplementary Fig. 1a). We could not detect any desired product and most of the starting material was recovered when *N*-methyl-substituted phenoxyacetamide (**1w**) was used under the S/Se-mediated reaction conditions, indicating the indispensable role of the N–H bond (Supplementary Fig. 1b). When compounds **1a** and $d_8$-**1a** were used as substrates under S-

**Table 3 Substrate scope of S-mediated reaction[a]**

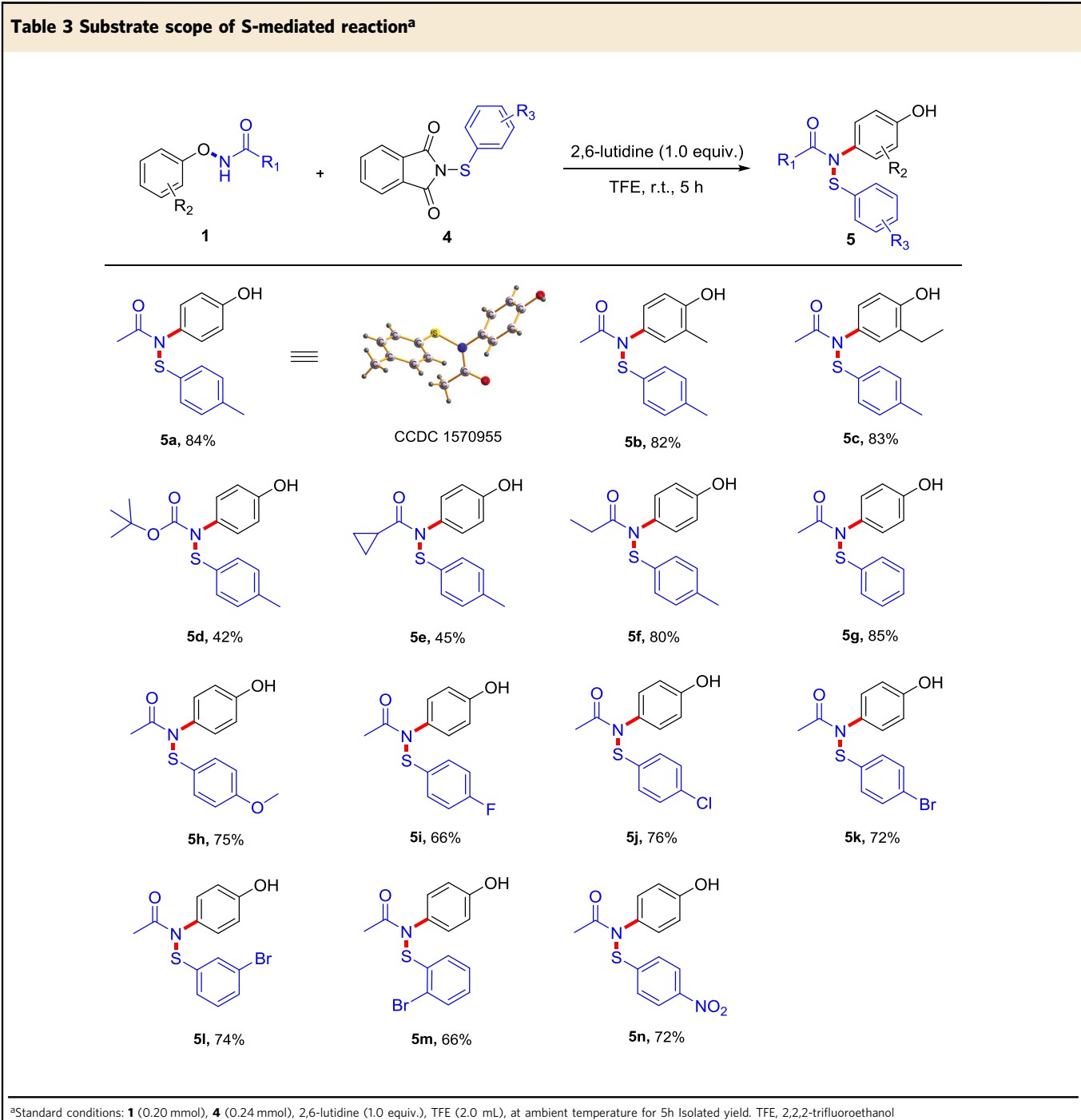

5a, 84% CCDC 1570955 5b, 82% 5c, 83%

5d, 42% 5e, 45% 5f, 80% 5g, 85%

5h, 75% 5i, 66% 5j, 76% 5k, 72%

5l, 74% 5m, 66% 5n, 72%

[a]Standard conditions: **1** (0.20 mmol), **4** (0.24 mmol), 2,6-lutidine (1.0 equiv.), TFE (2.0 mL), at ambient temperature for 5h Isolated yield. TFE, 2,2,2-trifluoroethanol

mediated reaction conditions, the HRMS data showed that the *para* amide transfer via an intramolecular pathway and the mixed acetamide migration products were not detected. In addition, a crossover experiment was carried out between equimolar amount of **1a** and $d_8$-**1a** under Se-catalysed reaction conditions in one reactor. Only the intramolecular amides transformation of phenols (**2a**, $d_7$-**2a**) were obtained (Supplementary Fig. 1c and Supplementary Fig. 13).

Based on the preliminary studies, the mechanism of this organoselenium-catalysed *para*-selective C–H bond amination is proposed in Fig. 2. The electrophilic Se species could react with the mildly basic *N*-phenoxyacetamide **1a** to give the Se–N

intermediate (**INT1-Se**) together with the release of one molecule of HBr. Then, the **INT1-Se** undergoes two successive [2,3]-sigmatropic rearrangements[44–49] to generate the *para*-amination intermediate (**INT3-Se**), which may readily react with HBr and then rearomatize to the desired product **2a** (for details see Supplementary Figs 4–11).

**DFT calculations.** We performed density functional theory (DFT) calculations to explore the mechanistic details for these S (and Se)-mediated *para*-selective nitrogen migration of *N*-aryloxyacetamides (Fig. 3). All calculations were carried out with the B3LYP functional[50,51], augmented with Grimmes D3

**Fig. 2** Proposed catalytic cycle of the organoselenium-catalysed *para*-amination of phenols. A plausible mechanism illustrating how **2a** is formed via two consecutive [2,3]-sigmatropic rearrangements

dispersion correction[52,53], which already proved to be a good choice for chalcogen-containing systems[54,55]. For S-mediated reaction, the reaction between *N*-phenoxyacetamide **1a** and *N*-phenylthiophthalimide **4g** was used as model reaction. The Gibbs energy profile is shown in Fig. 3a. First, the reaction of *N*-phenylthiophthalimide **4g** and **1a** generates the S–N intermediate **INT1-S**. Then, the [2,3]-sigmatropic rearrangement of **INT1-S** via **TS1-S** forms an *ortho*-S = N substituted dearomatized species **INT2-S**, with a barrier of 9.7 kcal mol$^{-1}$. Subsequently, the second [2,3]-sigmatropic rearrangement of **INT2-S** yields the *para*-amination intermediate **INT3-S** via **TS2-S** (with a barrier of 5.0 kcal mol$^{-1}$, see **path 1-S**). Finally, the aromatization of **INT3-S** generates the desired product **5g**. The whole process is exothermic by 43.9 kcal mol$^{-1}$, which indicates that the formation of **5g** is reasonable. However, the barrier for the regeneration of *N*-phenylthiophthalimide **4g** (via **TS$_{SN2}$**) is up to 32.3 kcal mol$^{-1}$, suggesting the turnover of **4g** is difficult even under basic condition. Therefore, for S-mediated reactions, a stoichiometric amount of *N*-phenylthiophthalimide is required (see Supplementary Fig. 8 for details). For the Se-catalysed reaction, the Gibbs energy profile of the reaction of **1a** and PhSeBr is shown in Fig. 3b. Although the reaction of PhSeBr and **1a** generating the Se–N intermediate **INT1-Se** is endothermic by 12.6 kcal mol$^{-1}$, **INT1-Se** may readily undergo a Se-centred [2,3]-sigmatropic rearrangement to generate an *ortho*-Se = N substituted dearomatized species (**INT2-Se**) via **TS1-Se**, with a barrier of 12.7 kcal mol$^{-1}$. Then, another *N*-centred [2,3]-sigmatropic rearrangement of **INT2-Se** forms *para*-amination intermediate **INT3-Se** via **TS2-Se** (with a barrier of 4.4 kcal mol$^{-1}$, see **path 1-Se**). Rearomatization of **INT3-**Se and regeneration of the active catalyst (PhSeBr) from **2a′** affords product **2a** readily with large Gibbs energy-driven forces (23.5 and 16.2 kcal mol$^{-1}$, respectively). In contrast to *N*-phenylthiophthalimide, the regeneration of PhSeBr is strongly exothermic by 14.7 kcal mol$^{-1}$ with a barrier of only 15.3 kcal mol$^{-1}$ (for details see Supplementary Fig. 11). Therefore, PhSeBr could be used as a catalyst. In addition to **path 1**, the direct rearomatization of **INT2** via **TS2′** to generate the *ortho*-S/Se = N substituted phenol (**INT2′**) is also possible (see **path 2-S** in Fig. 3a and **path 2-Se** in Fig. 3b). However, the activation barriers of **path 2** in these two systems are much higher than that of **path 1**. The calculated trends for the two reactions are consistent with the fact that no *ortho*-Se = N substituted phenol (or only small amount of *ortho*-S = N substituted phenol) was obtained for these two types of reactions. Therefore, **path 1** involving two successive [2,3]-sigmatropic rearrangements is mainly responsible for the two *para*-selective amination reactions (for details see Supplementary Figs 2–11 and Supplementary Data 1).

**Synthetic application**. To further explore the mild nature of our method, an HBT-substrate **1v** was subjected to the reaction condition in a mixed solvent of 95% PBS buffer and 5% 1,4-dioxane (Fig. 4a). The obtained product **2v** exhibits significant aggregation-induced emission behaviours[56–60]. The fluorescence intensity of the product increased gradually at 538 nm (Fig. 4b) in the reaction solution, accompanied by a dramatic change in emission colour from pale blue to bright yellow (Fig. 4c).

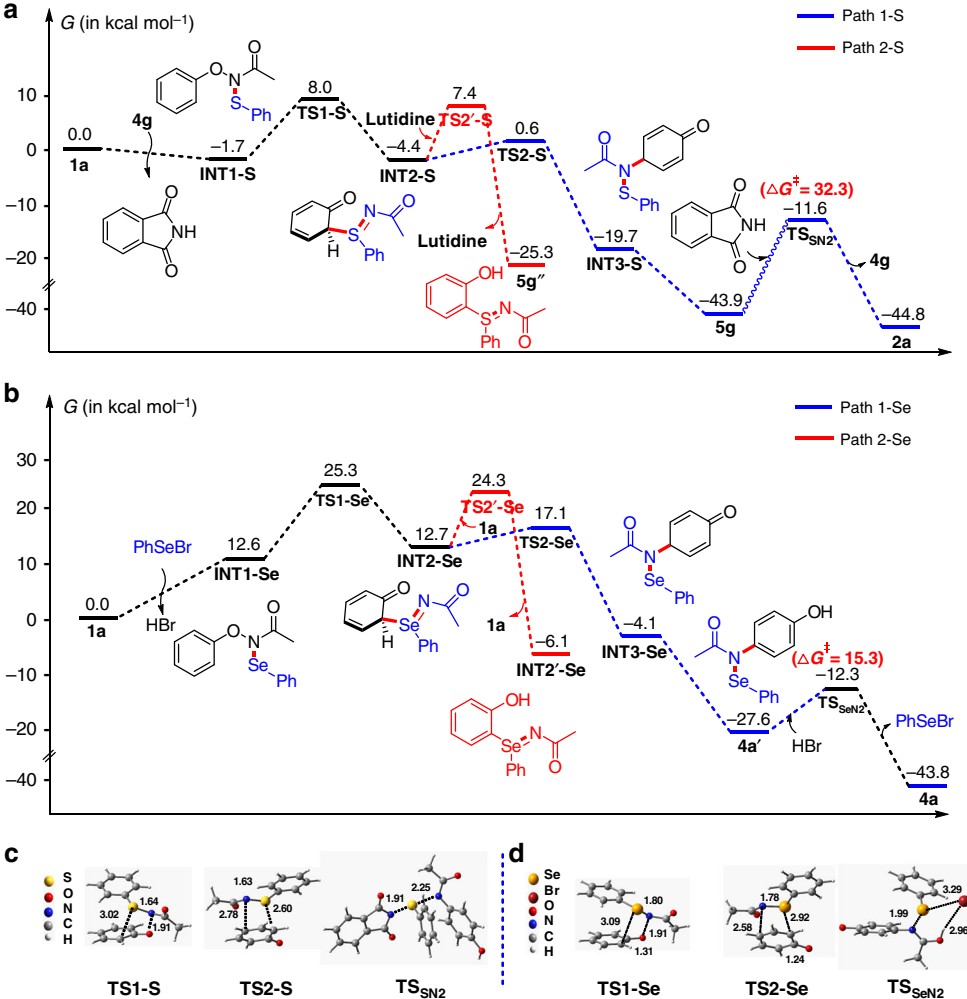

**Fig. 3** Computational studies on S (and Se)-mediated *para*-selective nitrogen migration of *N*-phenoxyacetamide (**1a**). **a** Computed Gibbs energy profile for S-mediated reaction (in TFE). **b** Computed Gibbs energy profile for Se-catalysed reaction (in 1,4-dioxane). **c** Transition states involved in S-mediated reaction. **d** Transition states involved in Se-catalysed condition

## Discussion

In summary, we discovered an organoselenium-catalysed *para*-amination of phenols or dienones under mild conditions. The methodology features a broad substrate scope and a high *para*-selectivity. More importantly, this work reveals a significant difference between the sulfenylation reagents and organoselenium reagents. While experimental and computational studies suggest that both the sulfur and selenium variants proceed through a double [2,3]-sigmatropic rearrangement, the sulfenylation reagents behave as coupling partners while organoselenium reagents can be employed catalytically. Because of the larger atomic radius of selenium compared to sulfur, selenium is more polarizable ("softer") than sulfur, allowing intrinsic selenium to be more nucleophilic and electrophilic[61,62]. Compared to sulfur, the larger hybridized orbitals of selenium results in weaker σ overlap[63]. So most bond strength of Se−X is weaker. The differences between sulfur and selenium developed here are reminiscent of their behaviours in biology. For example, the catalytic activity of the native enzyme dramatically reduces when the Sec residue in the type I ID enzyme was replaced by a cysteine (Cys) moiety[64,65]. We expect our present work to stimulate future studies of selenium as an alternative catalytic platform to transition metal-catalysed C–H amination reactions.

## Methods

**Materials**. For NMR spectra of compounds in this manuscript, see Supplementary Figs 14–73. For the crystallographic data of compound **2n** and **5a**, see Supplementary Fig. 12 and Supplementary Tables 3–15. For the representative experimental procedures and analytic data of compounds synthesized, see Supplementary Methods.

**Se-catalysed standard reaction conditions**. *N*-phenoxyamides (**1**) (0.20 mmol), PhSeBr (10 mol%), were weighed into a 10 mL tube, to which was added 1,4-dioxane (2.0 mL). The reaction vessel was stirred at room temperature for 8 h. Then the mixture was concentrated under vacuum and the residue was purified by column chromatography on silica gel with a gradient eluent of petroleum ether and ethyl acetate to afford the corresponding product **2** or **3**.

**S-mediated standard reaction conditions**. *N*-phenoxyacetamides (**1**) (0.20 mmol), *N*-substituted thiophthalimides (**4**) (0.24 mmol) and 2,6-lutidine (1.0 eq.) were weighed into a 10 mL tube, to which was added TFE (2.0 mL). The reaction vessel was stirred at room temperature for 5 h in air. The mixture was then concentrated under vacuum and the residue was purified by column chromatography on silica gel with a gradient eluent of petroleum ether and ethyl acetate to afford the corresponding product (**5**).

## Data availability

The X-ray crystallographic coordinates for structures reported in this study have been deposited at the Cambridge Crystallographic Data Centre (CCDC), under deposition number CCDC 1570955 and CCDC1549814. These data can be obtained free of charge from The Cambridge Crystallographic Data Centre via www.ccdc.cam.ac.uk/data_request/cif. The

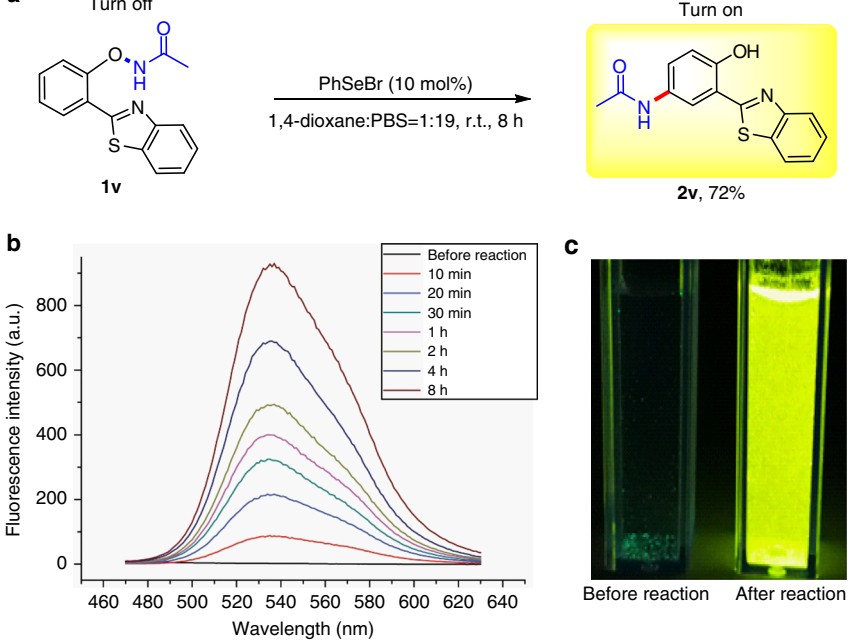

**Fig. 4** Application of the Se-catalysed reaction in aqueous conditions. **a** Conditions: **1v** (0.1 mmol), PhSeBr (10 mol%), DMSO/PBS buffer = 1:19 (4.0 mL); at ambient temperature for 8 h; the yield was isolated yield. **b** Fluorescence spectra of reaction in aqueous conditions, $\lambda_{ex} = 380$ nm. **c** Visual fluorescence of the reaction mixture under a 365 nm ultraviolet lamp

authors declare that all other data supporting the findings of this study are available within the article and Supplementary Information files, and also are available from the corresponding author upon reasonable request.

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

## Acknowledgements

Financial support was provided by the National Science Foundation of China (21622103, 21333004 and 21571098), the Natural Science Foundation of Jiangsu Province (BK20160022) and the Fundamental Research Funds for the Central Universities (No. 020514380117 and No. 020814380002).

## Author contributions

D.Y. and F.X. carried out the experimental work. The computational work was conducted by G.W.; D.Y. and G.W. prepared most of the manuscript and supporting information. J.Z., S.L., Z.S., Y.L. and W.-Y.S. guided the research.

## Additional information

**Competing interests:** The authors declare no competing interests.

