## [Peer Review File · Nature Communications]

Reviewer #1 (Remarks to the Author):

The manuscript presents an organoselenium-catalyzed para-amidation reaction of phenols. The reaction is proposed to proceed via a double [2,3]-sigmatropic rearrangement, and proves to be applicable to a broad scope of amide and phenol substrates. The manuscript also presents an analogous organosulfur-promoted para-amidation reaction of phenols, which leads to different amidated products bearing the N-S bond. The authors have performed control experiments and computational modeling to probe the reaction pathways, providing some insights on the possible intermediates in both N-S and N-Se mediated rearrangements. The studies are well performed with sufficient experimental information. The reported chemistry is interesting and novel, which is expected to find great use in organic chemistry.

Overall, the work is suitable for publication at Nature Communication. Yet before it can be accepted for publication, the manuscript should be revised to address the following questions/comments adequately.

Questions/comments:

1. The abstract states "A combination of experimental and computational studies were conducted to explain the fundamental difference between the N-Se bond and the N-S bond, which is reminiscent of their behaviors in biology." It remains unclear what "the fundamental difference" or "their behaviors in biology" refer to. There is no discussion regarding such statement in the manuscript, besides a single citation (ACS Chem Bio 2016). More detailed discussions are needed to justify such statement.

2. The authors refer the reactions in Table 1-3 as type I–III, yet there is no description how the type I, II or III reactions are defined. It will be helpful to include the intermediates in the reaction type to clarify the different pathways/products in Type I, II and III.

3. The section of Type I reactions is an important component of the chemistry presented in this manuscript. Yet most results are only listed in supplementary Table 2 in SI, and little information is provided in Figure 2 despite considerable amount of discussion in the manuscript. The authors are suggested to move these results in supplementary Table 2 in SI to the manuscript for clarity.

4. In SI, the HRMS calculated for new compounds are incorrect - the current values do not consider the positive charge. For example, the calculated HRMS for compound 3j, C₁₄H₁₃CINO₂S [M+H]⁺ should be 294.0350, not 294.0356. The authors need to check all the compounds and get them corrected accordingly.

5. In SI, the J coupling constant assignment for compound characterization should be checked for accuracy and consistency. Many of coupling constant assignments in the current version are found inappropriate, for example, the ¹H J coupling constants reported for compound 3m.

6. The presentation of the manuscript should be improved significantly, for clarity, conciseness and accuracy.

a. The authors should consider the choice of words through the manuscript.

For example, in the introduction, it states "there is an increasing demand for catalytic methods that encompass the mild and selective features of cross-coupling and employ unfunctionalized phenol substrates used in nitration." Although the background information explains the need for "catalytic methods that encompass the mild and selective features of cross-coupling and employ unfunctionalized phenol substrates". There is no clear justification for "the increasing demand". Here is another one: "Moreover, the developed reaction conditions were applicable to polycyclic substrate with moderate yield (4l, 54%)." Based on a single example, specifically naphthalenol derived amidated product 4l, the description aforementioned is an inappropriate overstatement for its applicability for polycyclic substrates.

b. Correct grammar errors and typos.

For example, TFE should be 2,2,2-Trifluoroethanol not 2,2,2-thifluoroethanol.

"Fig. 1 | A new metal-free methods to achieve 4-aminophenols" There are grammar errors.

"Screened of a variety of solvents, including MeOH, DMSO, THF, MeCN, EA and 1,4-dioxane (Supplementary Table 3, entries 8–13)" is not a complete sentence.

Reviewer #2 (Remarks to the Author):

Referee report for Zhao et al. Selenium Enables the Catalytic Intramolecular para-Amino Functionalization of Phenols" (ms. no. NCOMMS-18-10930)

This manuscript from Zhao et al. is a rewrite of a previous manuscript submitted to *Nature Catalysis* entitled "Organoselenium-Catalyzed para-Amidation of Phenols" (ms no. NATCATAL-17070154). Because the authors have chose to recast the manuscript and resubmit to the Nature group, without substantial change, I have reiterated my previous review below. The same conclusion applies here.

This manuscript from Zhao et al. is a follow up to their recently published article in *Nature Communications* (ref. 34) in which an identical transformation was carried out with a sulfonylating agent in place of a selenylating agent. In that study, the reaction of the aryl hydroxamic esters proceeded unsurprisingly to produce an ortho substituted sulfilimine. Herein, the identical process is executed, but the authors now find that the reaction does not stop at the ortho substituted selenimine, but rather continues on through a second [2,3] rearrangement to afford a para substituted phenol. The authors briefly survey selenylating reagents for their catalytic efficiency and then carry out a standard demonstration of scope. This section is followed by illustrations of routine transformations directed by the amide group as well as a few mechanistic studies that establish the intramolecular nature of the rearrangement. Finally a straightforward computational study substantiates the proposed mechanism and rules out a direct 1,5 shift.

Overall this is an interesting transformation but one that hardly rises to the stature of Nature Catalysis. The focus (unlike the previous article which emphasized the biocompatibility of the transformation) is primarily chemical, and as such is better suited for publication in the *Journal of Organic Chemistry* or *Chemistry European Journal*.

Reviewer #3 (Remarks to the Author):

This manuscript reports some interesting and publishable results, but I don't think the results are sufficiently significant to warrant publication in Nature Communications. I recommend publication in a more specialized journal.

The authors use DFT computations to support their observations and discussion of the experimental results, which is fine, but they provide no justification for their choice of the DFT method and basis sets employed. I suspect they have selected the DFT method on the basis of the publications by other authors on molecules containing selenium. A few key references should be added to the manuscript. The authors refer to free energies and Gibbs free energies. Thermodynamicists have long advocated using the term Gibbs energies and IUPAC has adopted their recommendation. Another trivial point: Angstroms should be written as angstroms to be consistent with SI conventions: joule or J, kelvin or K, etc.

Detailed Responses to Referee's Comments

Reviewers' comments:

Reviewer #1 (Remarks to the Author):

The manuscript presents an organoselenium-catalyzed *para*-amidation reaction of phenols. The reaction is proposed to proceed via a double [2,3]-sigmatropic rearrangement, and proves to be applicable to a broad scope of amide and phenol substrates. The manuscript also presents an analogous organosulfur-promoted *para*-amidation reaction of phenols, which leads to different amidated products bearing the N-S bond. The authors have performed control experiments and computational modeling to probe the reaction pathways, providing some insights on the possible intermediates in both N-S and N-Se mediated rearrangements. The studies are well performed with sufficient experimental information. The reported chemistry is interesting and novel, which is expected to find great use in organic chemistry.

Overall, the work is suitable for publication at *Nature Communication*. Yet before it can be accepted for publication, the manuscript should be revised to address the following questions/comments adequately.

Question 1:

1. The abstract states "A combination of experimental and computational studies were conducted to explain the fundamental difference between the N-Se bond and the N-S bond, which is reminiscent of their behaviors in biology." It remains unclear what "the fundamental difference" or "their behaviors in biology" refer to. There is no discussion regarding such statement in the manuscript, besides a single citation (ACS Chem Bio 2016). More detailed discussions are needed to justify such statement.

Response

We thank the reviewer for this valuable question. More details about the fundamental difference between selenium and sulphur as well as their different behaviors in biology

are discussed in our new edition. We have added several references in the revised manuscript.

Revisions Made

(Please refer to page 13, paragraph 13).

In summary, we discovered an organoselenium-catalysed *para*-amination of phenols or dienones under mild conditions. The methodology features a broad substrate scope and a high *para*-selectivity. More importantly, this work reveals significant a difference between the sulfenylation reagents and organoselenium reagents. While experimental and computational studies suggest that both the sulphur and selenium variants proceed through a double [2,3]-sigmatropic rearrangement, the sulfenylation reagents behave as coupling partners while organoselenium reagents can be employed catalytically. Since the larger atomic radius of selenium compared to sulphur, selenium are more polarizable (“softer”) than sulphur, allowing selenium intrinsic to be more nucleophilic and electrophilic^{61,62}. Compared to sulphur, the larger hybridized orbitals of selenium results in weaker σ overlap⁶³. So most bond strength of Se-X is weaker. The differences between sulphur and selenium developed here is reminiscent of their behaviors in biology; illustrates the potential of selenium to enable catalytic processes. Notably, type II and type III reactions reveal a novel organoselenium-catalyzed reaction in C-H functionlization with the unprecedented *para*-selectivity. For example, the catalytic activity of the native enzyme dramatically reduces when the Sec residue in the type I ID enzyme was replaced by a cysteine (Cys) moiety^{64,65}. We expect our present work to stimulate future studies of selenium as an alternative catalytic platform to transition metal-catalysed C-H amination reactions.

61. Steinmann, D., Nauser, T. & Koppenol, W. H. Selenium and sulphur in exchange reactions: a comparative study. *J. Org. Chem.* **75**, 6696-6699 (2010).

62. Trofymchuk, O. S., Zheng, Z., Kurogi, T., Mindiola, D. J. & Walsh, P. J. Selenolate anion as an organocatalyst: reactions and mechanistic studies. *Adv. Synth. Catal.* **360**, 1685-1692 (2018).

63. Reich, H. J. & Hondal, R. J. Why nature chose selenium. *ACS Chem. Biol.* **11**, 821-841 (2016).

64. Berry, M. J., Kieffer, J. D., Harney, J. W. & Larsen, P. R. Selenocysteine confers the biochemical

properties characteristic of the type I iodothyronine deiodinase. *J. Biol. Chem.* **266**,14155-14158 (1991).

65. Larsen, P. R. & Berry, M. J. Nutritional and hormonal regulation of thyroid hormone deiodinases. *Annu. Rev. Nutr.* **15**, 323-352 (1995).

Question 2:

2. The authors refer the reactions in Table 1-3 as type I-III, yet there is no description how the type I, II or III reactions are defined. It will be helpful to include the intermediates in the reaction type to clarify the different pathways/products in Type I, II and III.

Response

We thank the reviewer for this valuable advice. We have renamed the reactions in Table 1-3 as Se-catalysed (referred to Type II and III) and S-mediated (referred to Type I) reactions. Details have been displayed in the revised manuscript.

Question 3:

3. The section of Type I reactions is an important component of the chemistry presented in this manuscript. Yet most results are only listed in supplementary Table 2 in SI, and little information is provided in Figure 2 despite considerable amount of discussion in the manuscript. The authors are suggested to move these results in supplementary Table 2 in SI to the manuscript for clarity.

Response

We thank the reviewer for this valuable advice. We have move the Supplementary Table 2 in SI to the new manuscript as Table 3.

Revisions Made

(Please refer to page 7, Table 3).

Table 3 | Substrate scope of S-mediated reaction^a

^aStandard conditions: **1** (0.20 mmol), **4** (0.24 mmol), 2,6-lutidine (1.0 eq.), TFE (2.0 mL), at ambient temperature for 5 h. TFE, 2,2,2-thi fluoroethanol, 2,2,2-trifluoroethanol. Isolated yield.

Question 4:

4. In SI, the HRMS calculated for new compounds are incorrect - the current values do not consider the positive charge. For example, the calculated HRMS for compound 3j, C₁₄H₁₃ClNO₂S [M+H]⁺ should be 294.0350, not 294.0356. The authors need to check

all the compounds and get them corrected accordingly.

Response

We are very sorry for our negligence of positive charges. We have checked all the compounds and corrected accordingly.

Revisions Made

(Please refer to Pages 6-21 in SI)

Take compound **3j5j** for example:

N-((4-chlorophenyl)thio)-N-(4-hydroxyphenyl)acetamide, 3j5j, white solid, 46.3 mg, 0.158 mmol, yield: 79%

¹H NMR (400 MHz, DMSO): δ 9.67 (s, 1H), 7.46–7.42 (m, 2H), 7.35–7.30 (m, 2H), 7.10 (d, $J = 8.6$ Hz, 2H), ~~6.73 (d, $J = 8.8$ Hz, 2H)~~6.75–6.71 (m, 2H), 2.11 (s, 3H); **¹³C NMR (101 MHz, DMSO):** δ 173.66, 157.28, 137.28, 136.98, 131.72, 129.66, 128.59, 126.85, 116.12, 22.96; **HRMS (ESI) calculated for C₁₄H₁₃ClNO₂S [M+H]⁺:** ~~294.0356~~294.0350; Found: 294.0356.

Question 5:

5. In SI, the J coupling constant assignment for compound characterization should be checked for accuracy and consistency. Many of coupling constant assignments in the current version are found inappropriate, for example, the ¹H J coupling constants reported for compound 3m.

Response

We were really sorry for this mistake. We have checked all the J coupling constants of the compounds and corrected them for accuracy.

Revisions Made

(Please refer to Pages 6-21 in SI)

Take compound **3m5m** for example:

N-((2-bromophenyl)thio)-N-(4-hydroxyphenyl)acetamide, 3m5m, white solid, 44.5 mg, 0.132 mmol, yield: 66%

¹H NMR (500 MHz, DMSO): δ 9.72 (s, 1H), 7.55 (dd, $J = 7.9, 0.8$ Hz, 1H), 7.45 (~~t~~, $J = 7.5$ Hz m, 1H), 7.36 (d, $J = 7.4$ Hz, 1H), 7.25 (d, $J = 8.1$ Hz, 2H), 7.12 (~~dd~~, $J = 10.9, 4.3$ Hz m, 1H), 6.74 (d, $J = 8.7$ Hz, 2H), 2.12 (s, 3H); **¹³C NMR (126 MHz, DMSO):** δ 173.34, 157.37, 138.59, 136.73, 133.18, 128.96, 128.63, 127.66, 124.16, 116.17, 115.54, 22.88; **HRMS (ESI)** calculated for C₁₄H₁₂BrNNaO₂S [M+Na]⁺: 359.9670359.9664; Found: 359.9668.

Question 6:

6. The presentation of the manuscript should be improved significantly, for clarity, conciseness and accuracy.

a. The authors should consider the choice of words through the manuscript.

For example, in the introduction, it states “there is an increasing demand for catalytic methods that encompass the mild and selective features of cross-coupling and employ unfunctionalized phenol substrates used in nitration.” Although the background information explains the need for “catalytic methods that encompass the mild and selective features of cross-coupling and employ unfunctionalized phenol substrates”. There is no clear justification for “the increasing demand”.

Here is another one: “Moreover, the developed reaction conditions were applicable to polycyclic substrate with moderate yield (4l, 54%).” Based on a single example, specifically naphthalenol derived amidated product 4l, the description aforementioned is an inappropriate overstatement for its applicability for polycyclic substrates.

b. Correct grammar errors and typos.

For example, TFE should be 2,2,2-Trifluoroethanol not 2,2,2-thifluoroethanol.

“Fig. 1 | A new metal-free methods to achieve 4-aminophenols” There are grammar errors.

“Screened of a variety of solvents, including MeOH, DMSO, THF, MeCN, EA and 1,4-dioxane (Supplementary Table 3, entries 8–13)” is not a complete sentence.

Response

We thank the reviewer for his/her constructive advice that have helped us to improve our manuscript. We have rewritten the introduction and corrected the mistakes. The following revision is provided.

Revisions Made

(Please refer to Page 2, paragraphs 1, 2; Page 3, Fig. 1)

Selenium is an essential biological trace element discovered by the Jöns Jacob Berzelius in 1818¹. The selenium analogue of cysteine, known as selenocysteine²⁻⁴ (Sec), is the main biological form of selenium. The most studied selenoenzyme glutathione peroxidases (GPx) have a Sec residue in its active site which is responsible for decomposing hydroperoxides (Fig. 1a)^{5,6}. Besides, the flavin-containing redox enzyme thioredoxin reductase (TrxR)⁷⁻⁹ and the deiodinating enzyme iodothyronine deiodinase (ID)^{10,11} represent other key selenium-containing enzymes in biocatalysis.

Selenium-containing small molecules, such as ebselen and its analogues, have also exhibited important antioxidant activity as GPx mimics¹²⁻¹⁵. Organoselenium-catalysed reactions have been widely employed in a number of different reactions¹⁶⁻¹⁸, and substantial progress have been made by Breder¹⁹⁻²¹, Wirth²²⁻²⁴, Denmark^{25,26}, Yeung²⁷ and Zhao²⁸⁻³¹ in recent years. Notably, selenium has emerged as appropriate alternatives to precious metals as catalysts for the construction of C–N bonds³²⁻³⁴. Breder *et al.*

Fig. 1 | Selected biological reaction and organic reactions catalysed by selenium. (a) Proposed catalytic cycle of glutathione peroxidase (GPx) for the reduction of hydroperoxides in biology. **(b)** Previous reports on organoselenium-catalysed amination of alkenes. GS⁻, glutathione. **(c)** Our double [2,3]-sigmatropic rearrangement to achieve *para*-amination of phenols.

discovered an elegant selenium-catalysed amination of allyl and vinyl using *N*-fluorobenzenesulfonimide (NFSI) as oxidant and nitrogen source³⁵. Furthermore, Zhao *et al.* accomplished a powerful pyridination of 1,3-dienes using (BnSe)₂ as a catalyst³⁶ (Fig. 1b). However, no selenium-catalysed processes for the functionalisation of aromatic compounds have been developed. One challenge might be the electrophilic selenium catalysts (ESC) react with the aryl rings directly, leading to the deactivation of catalyst^{37,38}. We thought that a more nucleophilic site, to accommodate with selenium

catalyst temporarily, might be helpful for competing with the deactivation. We herein report a strategy to first form an intermediate with an adjacent, redox versatile Se–N bond which undergoes two successive sigmatropic rearrangements to generate the *para*-amination product and regenerate the selenium catalyst (Fig. 1c).

(Please refer to Page 4, paragraph 3)

~~Screened of a variety of solvents, including MeOH, DMSO, THF, MeCN, EA and 1,4-dioxane (Supplementary Table 3, entries 8–13). It revealed that 1,4-dioxane was the ideal solvent providing 93% yield (NMR) and 90% isolated yield of the desired product. Screening of a variety of solvents (including MeOH, DMSO, THF, MeCN, EA) indicated that 1,4-dioxane was the best solvent (93% NMR yield and 90% isolated yield of the desired product, Supplementary Table 1, entries 8–13).~~

(Please refer to Page 4, paragraph 4)

~~Moreover, the developed reaction conditions were applicable to polycyclic substrate with moderate yield (**4I**, 54%).~~

The reaction condition was applicable to yield aminated naphthol in 54% yield (**2I**).

(Please refer to Page 7)

“Standard conditions: **1** (0.20 mmol), ~~**2**~~ **4** (0.24 mmol), 2,6-lutidine (1.0 eq.), TFE (2.0 mL), at ambient temperature for 5 h. TFE, ~~2,2,2-trifluoroethanol~~ 2,2,2-trifluoroethanol. Isolated yield.

Reviewer #2 (Remarks to the Author):

This manuscript from Zhao et al. is a rewrite of a previous manuscript submitted to Nature Catalysis entitled “Organoselenium-Catalyzed *para*-Amidation of Phenols” (msno. NATCATAL-17070154). Because the authors have chose to recast the manuscript and resubmit to the Nature group, without substantial change, I have reiterated my previous review below. The same conclusion applies here.

This manuscript from Zhao et al. is a follow up to their recently published article in

Nature Communications (ref. 34) in which an identical transformation was carried out with a sulfonylating agent in place of a selenylating agent. In that study, the reaction of the aryl hydroxamic esters proceeded unsurprisingly to produce an ortho substituted sulfilimine. Herein, the identical process is executed, but the authors now find that the reaction does not stop at the ortho substituted selenimine, but rather continues on through a second [2,3] rearrangement to afford a para substituted phenol. The authors briefly survey selenylating reagents for their catalytic efficiency and then carry out a standard demonstration of scope. This section is followed by illustrations of routine transformations directed by the amide group as well as a few mechanistic studies that establish the intramolecular nature of the rearrangement. Finally a straightforward computational study substantiates the proposed mechanism and rules out a direct 1,5 shift. Overall this is an interesting transformation but one that hardly rises to the stature of Nature Catalysis. The focus (unlike the previous article which emphasized the biocompatibility of the transformation) is primarily chemical, and as such is better suited for publication in the Journal of Organic Chemistry or Chemistry European Journal.

Response

We thank the reviewer for encouraging comments on our manuscript. We have rewritten the introduction which emphasises the significance of our work for the development of selenium catalysis on aromatic compounds.

Besides, we also demonstrated the biocompatibility of the *para*-amination reaction by generating an AIEgen 2-(2'-hydroxyphenyl)benzothiazole (HBT) product in a fluorogenic fashion in a PBS buffer (Fig. 4).

Revisions Made

(Please refer to Page 2, paragraphs 1, 2; Page 3, Fig. 1)

Selenium is an essential biological trace element discovered by the Jöns Jacob Berzelius in 1818¹. The selenium analogue of cysteine, known as selenocysteine²⁻⁴ (Sec), is the main biological form of selenium. The most studied selenoenzyme glutathione peroxidases (GPx) have a Sec residue in its active site which is responsible for

decomposing hydroperoxides (Fig. 1a)^{5,6}. Besides, the flavin-containing redox enzyme

Fig. 1 | Selected biological reaction and organic reactions catalysed by selenium. (a) Proposed catalytic cycle of glutathione peroxidase (GPx) for the reduction of hydroperoxides in biology. **(b)** Previous reports on organoselenium-catalysed amidation of alkenes. GS⁻, glutathione. **(c)** Our double [2,3]-sigmatropic rearrangement to achieve *para*-amidation of phenols.

thioredoxin reductase (TrxR)⁷⁻⁹ and the deiodinating enzyme iodothyronine deiodinase (ID)^{10,11} represent other key selenium-containing enzymes in biocatalysis.

Selenium-containing small molecules, such as ebselen and its analogues, have also exhibited important antioxidant activity as GPx mimics¹²⁻¹⁵. Organoselenium-catalysed reactions have been widely employed in a number of different reactions¹⁶⁻¹⁸, and substantial progress have been made by Breder¹⁹⁻²¹, Wirth²²⁻²⁴, Denmark^{25,26}, Yeung²⁷ and

Zhao²⁸⁻³¹ in recent years. Notably, selenium has emerged as appropriate alternatives to precious metals as catalysts for the construction of C–N bonds³²⁻³⁴. Breder *et al.* discovered an elegant selenium-catalysed amination of allyl and vinyl using *N*-fluorobenzenesulfonimide (NFSI) as oxidant and nitrogen source³⁵. Furthermore, Zhao *et al.* accomplished a powerful pyridination of 1,3-dienes using (BnSe)₂ as a catalyst³⁶ (Fig. 1b). However, no selenium-catalysed processes for the functionalisation of aromatic compounds have been developed. One challenge might be the electrophilic selenium catalysts (ESC) react with the aryl rings directly, leading to the deactivation of catalyst^{37,38}. We thought that a more nucleophilic site, to accommodate with selenium catalyst temporarily, might be helpful for competing with the deactivation. We herein report a strategy to first form an intermediate with an adjacent, redox versatile Se–N bond which undergoes two successive sigmatropic rearrangements to generate the *para*-amination product and regenerate the selenium catalyst (Fig. 1c).

(Please refer to Page 12, Fig. 4; Page 13, paragraph 12)

Fig.4 | Application of the Se-catalysed reaction in aqueous conditions. (a) Conditions: **1v (0.1 mmol), PhSeBr (10 mol%), DMSO/PBS buffer = 1:19 (4.0 mL); at ambient temperature for 8 h; the yield was isolated yield. (b) Fluorescence spectra of reaction in aqueous conditions, $\lambda_{\text{ex}} = 380 \text{ nm}$. (c) Visual fluorescence of the reaction mixture under a 365 nm ultraviolet lamp.**

Synthetic application. To further explore the biocompatibility of our method, a HBT-substrate **1v** was subjected to the reaction condition in a mixed solvent of 95% PBS buffer and 5% 1,4-dioxane (Fig. 4a). The obtained product **2v** exhibits significant aggregation-induced emission (AIE) behaviors⁵⁶⁻⁶⁰. The fluorescence intensity of the product increased gradually at 538 nm (Fig. 4b) in the reaction solution, accompanied by a dramatic change in emission colour from pale blue to bright yellow (Fig. 4c).

Reviewer #3 (Remarks to the Author):

This manuscript reports some interesting and publishable results, but I don't think the results are sufficiently significant to warrant publication in Nature Communications. I recommend publication in a more specialized journal.

The authors use DFT computations to support their observations and discussion of the experimental results, which is fine, but they provide no justification for their choice of the DFT method and basis sets employed. I suspect they have selected the DFT method on the basis of the publications by other authors on molecules containing selenium. A few key references should be added to the manuscript. The authors refer to free energies and Gibbs free energies. Thermodynamicists have long advocated using the term Gibbs energies and IUPAC has adopted their recommendation. Another trivial point: Angstroms should be written as angstroms to be consistent with SI conventions: joule or J, kelvin or K, etc.

Question 1:

The authors use DFT computations to support their observations and discussion of the experimental results, which is fine, but they provide no justification for their choice of the DFT method and basis sets employed. I suspect they have selected the DFT method on

the basis of the publications by other authors on molecules containing selenium. A few key references should be added to the manuscript.

Response

We thank the reviewer for this excellent suggestion. The use of B3LYP-D3 functional in this work is mainly due to its good performance in describing chalcogen-containing systems. We have added discussions in the revised manuscript, and the related references have been cited as refs No. 50-55 in revised manuscript.

Question 2:

The authors refer to free energies and Gibbs free energies. Thermodynamicists have long advocated using the term Gibbs energies and IUPAC has adopted their recommendation. Another trivial point: Angstroms should be written as angstroms to be consistent with SI conventions: joule or J, kelvin or K, etc.

Response

We thank the reviewer for this excellent suggestion. We have changed all of “free energies” and “Gibbs free energies” to “Gibbs energies” in the revised manuscript and SI. Besides, “Angstroms” have been written as “Å” as suggested in the SI.

Revisions Made

(Please refer to Page 10, paragraph 11; Page 11, Fig. 3)

DFT calculations. We performed density functional theory (DFT) calculations to explore the mechanistic details for these S (and Se)-mediated *para*-selective nitrogen migration of *N*-aryloxyacetamides (Fig. 53). All calculations were carried out with the B3LYP functional^{50,51}, augmented with Grimmes D3 dispersion correction^{52,53}, which already proved to a good choice for chalcogen containing systems^{54,55}. For reaction-type-I S-mediated reaction, the reaction between *N*-phenoxyacetamide **1a** and *N*-phenylthiophthalimide **2g 4g** was used as model reaction. The free-energy-Gibbs energy profile is shown in Fig. 53a. First, the reaction of *N*-phenylthiophthalimide **4g 2g** and **1a** generates the S–N intermediate **INT1-S**. Then, the [2,3]-sigmatropic rearrangement of

INT1-S via TS1-S forms a *ortho*-S=N substituted dearomatized species INT2-S, with a

Fig. 53 | Computational studies on S (and Se)-mediated *para*-selective nitrogen migration of *N*-phenoxyacetamide (**1a**). (a) Computed free energy Gibbs energy profile for reaction type I S-mediated reaction (in TFE). (b) Computed free energy Gibbs energy profile for reaction type II Se-catalysed reaction (in 1,4-dioxane). (c) Transition states involved in reaction type I S-mediated reaction. (d) Transition states involved in reaction type II Se-catalysed reaction.

barrier of 9.7 kcal mol⁻¹. Subsequently, the second [2,3]-sigmatropic rearrangement of INT2-S yields the *para*-amination intermediate INT3-S via TS2-S (with a barrier of 5.0 kcal mol⁻¹, see **path 1-S**). Finally, the aromatization of INT3-S generates the desired product **5g-3g**. The whole process is exothermic by 43.9 kcal mol⁻¹, which indicates that the formation of **5g-3g** is reasonable. However, the barrier for the regeneration of *N*-phenylthiophthalimide **4g-2g** (via TS_{SN2}) is up to 32.3 kcal mol⁻¹, suggesting the turnover of **4g-2g** is difficult even under basic condition. Therefore, for **reaction-type-I S-mediated reactions**, a stoichiometric amount of *N*-phenylthiophthalimide is required (see Supplementary Fig. 8 for details). For **reaction-type-II the Se-catalysed reaction**, the **free energy Gibbs energy** profile of the reaction of **1a** and PhSeBr is shown in Fig. **53b**. Although the reaction of PhSeBr and **1a** generating the Se-N intermediate INT1-Se is endothermic by 12.6 kcal mol⁻¹, INT1-Se may readily undergo a Se-centered [2,3]-sigmatropic rearrangement to generate an *ortho*-Se=N substituted dearomatized species (INT2-Se) via TS1-Se, with a barrier of 12.7 kcal mol⁻¹. Then, another *N*-centered [2,3]-sigmatropic rearrangement of INT2-Se forms *para*-amination intermediate INT3-Se via TS2-Se (with a barrier of 4.4 kcal mol⁻¹, see **path 1-Se**). Rearomatization of INT3-Se and regeneration of the active catalyst (PhSeBr) from **4a-2a'** affords product **4a2a** readily with large **free energy Gibbs energy** driven forces (23.5 and 16.2 kcal mol⁻¹, respectively). In contrast to *N*-phenylthiophthalimide, the regeneration of PhSeBr is strongly exothermic by 14.7 kcal mol⁻¹ with a barrier of only 15.3 kcal mol⁻¹ (for details see Supplementary Fig. 11). Therefore, PhSeBr could be used as a catalyst. In addition to the **path 1**, the direct rearomatization of INT2 via TS2' to generate the *ortho*-S/Se=N substituted phenol (INT2') is also possible (see **path 2-S** in Fig. **53a** and **path 2-Se** in Fig. **53b**). However, the activation barriers of **path 2** in these two systems are much higher than that of **path 1**. The calculated trends for the two reactions are consistent with the fact that no *ortho*-Se=N substituted phenol (or only small amount of *ortho*-S=N substituted phenol) was obtained for these two types of reactions. Therefore, **path 1** involving two successive [2,3]-sigmatropic rearrangements is mainly responsible for the two *para*-selective amination reactions (for details see Supplementary Figs 2-11).

98, 5648-5652 (1993).

51. Lee, C., Yang, W. & Parr, R. G. Development of the Colle-Salvetti correlationenergy formula into a functional of the electron density. *Phys. Rev. B Condens. Matter* **37**, 785–789 (1988).

52. Grimme, S., Antony, J., Ehrlich, S. & Krieg, H. A consistent and accurate ab initio parametrization of density functional dispersion correction (DFT-D) for the 94 elements H-Pu. *J. Chem. Phys.* **132**, 154104-154119 (2010).

53. Grimme, S., Hansen, A., Brandenburg, J. G., & Bannwarth, C. Dispersion-corrected mean-field electronic structure methods. *Chem. Rev.* **116**, 5105-5154 (2016).

54. Bleiholder, C., Werz, D. B., Köppel, H. & Gleiter, R. Theoretical investigations on chalcogen-chalcogen interactions: what makes these nonbonded interactions bonding? *J. Am. Chem. Soc.* **128**, 2666-2674 (2006).

55. Gleiter, R., Haberhauer, G., Werz, D.B.; Rominger, F., Bleiholder, C. From noncovalent chalcogen-chalcogen interactions to supramolecular aggregates: experiments and calculations. *Chem. Rev.* **118**, 2010-2041 (2018).

Reviewer #1 (Remarks to the Author):

The revised manuscript by Zhao and co-authors has made considerable improvements over the original version. The revision has addressed most of the questions/comments raised by this reviewer in the previous report. The revision adds a new example of the Se-catalyzed amidation reaction for the formation of an AIE gen 2-(2'-hydroxyphenyl)benzothiazole (HBT) product. This is a nice demonstration of its compatibility under the aqueous conditions (95% PBS buffer and 5% 1,4-dioxane). Yet it is not demonstration of the "biocompatibility", as significantly more experiments are required to demonstrate biocompatibility than the use of PBS buffer. The authors should remove "biocompatible" in the title and the "biocompatibility" through the manuscript if no further experiments are provided. There are still some typos to be corrected through the manuscript. Overall, the reviewer recommends its acceptance for publication after these minor revisions.

Note that this review report does not provide any comments on the computational studies, which is out of expertise of this reviewer.